# Immune Responses against SARS-CoV-2—Questions and Experiences

**DOI:** 10.3390/biomedicines9101342

**Published:** 2021-09-28

**Authors:** Harald Mangge, Markus Kneihsl, Wolfgang Schnedl, Gerald Sendlhofer, Francesco Curcio, Rossana Domenis

**Affiliations:** 1Clinical Institute of Medical and Chemical Laboratory Diagnostics, Medical University of Graz, 8036 Graz, Austria; markus.kneihsl@medunigraz.at (M.K.); w.schnedl@dr-schnedl.at (W.S.); 2Department of Neurology, Medical University of Graz, 8036 Graz, Austria; 3Practice for General Internal Medicine, Dr. Theodor Körnerstrasse 19b, A-8600 Bruck, Austria; 4Executive Department for Quality and Risk Management, University Hospital Graz, 8036 Graz, Austria; Gerald.Sendlhofer@uniklinikum.kages.at; 5Research Unit for Safety in Health, Department of Surgery, Medical University of Graz, 8036 Graz, Austria; 6Department of Medical Area, University of Udine, 33100 Udine, Italy; francesco.curcio@asufc.sanita.fvg.it (F.C.); rossana.domenis@uniud.it (R.D.)

**Keywords:** SARS-CoV-2, inflammation, T and B cell responses, immunity, development of the COVID-19 pandemic

## Abstract

Understanding immune reactivity against SARS-CoV-2 is essential for coping with the COVID-19 pandemic. Herein, we discuss experiences and open questions about the complex immune responses to SARS-CoV-2. Some people react excellently without experiencing any clinical symptoms, they do not get sick, and they do not pass the virus on to anyone else (“sterilizing” immunity). Others produce antibodies and do not get COVID-19 but transmit the virus to others (“protective” immunity). Some people get sick but recover. A varying percentage develops respiratory failure, systemic symptoms, clotting disorders, cytokine storms, or multi-organ failure; they subsequently decease. Some develop long COVID, a new pathologic entity similar to fatigue syndrome or autoimmunity. In reality, COVID-19 is considered more of a systemic immune–vascular disease than a pulmonic disease, involving many tissues and the central nervous system. To fully comprehend the complex clinical manifestations, a profound understanding of the immune responses to SARS-CoV-2 is a good way to improve clinical management of COVID-19. Although neutralizing antibodies are an established approach to recognize an immune status, cellular immunity plays at least an equivalent or an even more important role. However, reliable methods to estimate the SARS-CoV-2-specific T cell capacity are not available for clinical routines. This deficit is important because an unknown percentage of people may exist with good memory T cell responsibility but a low number of or completely lacking peripheral antibodies against SARS-CoV-2. Apart from natural immune responses, vaccination against SARS-CoV-2 turned out to be very effective and much safer than naturally acquired immunity. Nevertheless, besides unwanted side effects of the currently available vector and mRNA preparations, concerns remain whether these vaccines will be strong enough to defeat the pandemic. Altogether, herein we discuss important questions, and try to give answers based on the current knowledge and preliminary data from our laboratories.

## 1. Introduction

The innate and the adaptive immune system cooperate and coincide for an effective immune response to a severe acute respiratory syndrome infection of coronavirus 2 (SARS-CoV-2). In the best-case scenario, the person becomes immune against the pathogen without clinical symptoms. However, in the worst-case scenario, deficiency or over-activation of the immune response could lead to severe illnesses accompanied by cytokine storms, respiratory failure, and death. SARS-CoV-2 uses a spike (S) protein to infect human cells via the angiotensin-2 (ACE2) receptor. Observational studies of SARS-CoV-2 infection and vaccine trials [1,2] provide evidence that robust neutralizing antibody titers and virus-specific T cell responses against the S protein effectively protect against severe COVID-19 [3,4]. Thus, this protein became a main target with which to stimulate the immune system by vaccination. The favorable risk–benefit ratio of vaccination certainly make this intervention an effective medicinal approach. Nevertheless, the worldwide vaccination campaign has not defeated the pandemic so far. In the following paper we will discuss questions that are still open concerning the immune response to SARS-CoV-2 infection, vaccination, and recently emerging adverse reactions based on unwanted effects of the vaccine-related immune activation [5,6,7,8,9].

## 2. Questions Addressing the Human Immune Answer to Infection with SARS-CoV-2

### 2.1. Mild Versus Severe Disease Course: How Different Is the Immune Response?

A pronounced initial immune response and higher SARS-CoV-2 copy numbers at diagnosis usually correlate with the severity of disease [8]. In fatal cases, the immune response itself is pathogenic. This can occur through the production of antibodies [10] that damage tissues [11,12], or through inducing excessive inflammation, resulting in a cytokine storm [13,14]. The immunological measurements of the immune response to SARS-CoV-2 are concentrated so far on total IgG antibody levels, with some groups also reporting IgA, IgM, and neutralizing antibody levels. Only occasionally, memory B and T cell levels are also included in the assessments. This is a considerable deficit because the cellular immune capacity is an important element with which to control SARS-CoV-2. Better immunological metrics must be developed, and reliable biomarkers to diagnose earlier courses with a subsequent damaging immune response are missing as of now. Figure 1 outlines in detail the important facets of the immune response found in mild compared to severe disease. The clinical manifestation of COVID-19 frequently shows smooth transitions between these two extreme phenotypes. It remains unknown which main determinants are responsible for a shift to a bad course. Advanced age [15], cardiovascular disease (CVD), and obesity [16] can also include the genetic factors that could be involved in causing severe COVID-19 [15]. Nevertheless, a minor subset of young and middle-aged individuals also develop massive inflammatory responses and need intensive care treatment with mechanical ventilation [14]. Several studies have tried to find out mediators which indicate a switch to a hyper-inflammatory reaction pattern [13,17,18,19,20]. Genetic predisposition may play an important role in this context [21]. Furthermore, late COVID immune responses are an emerging problem. Some individuals with a mild initial course of COVID-19 suffer from chronic symptoms with a duration of >2 months after the initial infection, called long COVID syndrome [22,23]. The symptoms resemble other postinfectious syndromes following outbreaks of Ebola [24] and chikungunya [25]. There is some overlap with myalgic encephalomyelitis [26], with dysregulated autonomic nervous system components and perturbed immune parameters (https://www.biorxiv.org/content/10.1101/2020.02.20.958249v2, accessed on 28 February 2020). The currently extensive impact of COVID-19 offers a great chance for science to improve the understanding of this long-lasting postinfection syndrome, which is probably based on a disturbed neuro-endocrino-immunological axis.

#### 2.1.1. Grey Area

In the early phase of COVID-19, activated peripheral blood CD4^+^ and CD8^+^ T lymphocyte fractions arise in the circulation. CD4^+^ T cells show an increased level of IL-2 and IL-17 production. T regulatory cell (Tregs), neutrophil, and classic monocyte levels are also increased. Numbers of B cells and monocytes remain unchanged and increase later on in the course. The CD4^+^ and CD8^+^ T effector and memory lymphocytes are activated (CD38^+^, HLA-DR^+^) and show exhaustion-associated markers (PD-1, CD54). The chemokine receptor expression pattern shows decreased CCR6, CCR4, CXCR3, and CXCR4 antigens.

#### 2.1.2. Blue Area

A subsequent favorable clinical course is associated with increased numbers of circulating dendritic cells, recovery of T cell numbers by polyfunctional and follicular helper T cell fractions, decreased neutrophils, and an increased number of intermediate HLA-DR-expressing monocytes. The humoral antibody response starts via activated and expanded B cells. 

#### 2.1.3. Red Area

Bad disease courses show decreased levels of dendritic and natural killer cell fractions and increased levels of immature neutrophils. Monocytes downregulate HLA-DR expression and upregulate IL-1β, TNF, and interferon-stimulated genes (ISGs). Although low in absolute numbers, CD8^+^ T cells upregulate granzyme B and perforin secretion. While the number of plasma blasts increases, specific antibodies are consistently produced. The main point is that myeloid cells show pro-inflammatory behavior with an egress of immature neutrophils from the bone marrow and an enhanced formation of neutrophil extracellular nets (NETs). NET formation is an important contributor to the development of organ and endothelial injury in conjunction with activation of the complement and coagulation cascades and a so-called cytokine storm.

### 2.2. How Is an Immune Response after Infection with the Virus Currently Measured?

Neutralizing antibodies to SARS-CoV-2 are measured because they detect unique components of the virus. The virus has four main structural proteins: the spike (S) glycoprotein, the small envelope (E) glycoprotein, the membrane (M) glycoprotein, and the nucleocapsid (N) protein. Memory T cells and memory B cells also recognize these components, but they are difficult to measure. Notably, the presence of antibodies does not always predict the presence of specific T cells or memory B cells [28,29]. Interestingly, the CD4^+^ T cell response equally targets several SARS-CoV-2 proteins, whereas the CD8^+^ T cell response in turn targets the nucleoprotein. This observation indicates the importance of including the nucleoprotein as a potential antigen target in future vaccines [30]. Nevertheless, SARS-CoV-2 S and N peptides are not potently immunogenic, and none of the single peptides could universally induce robust T cell responses, suggesting the necessity of using a multi-epitope strategy for COVID-19 vaccine design [31].

### 2.3. How Do Innate and Adaptive Immune Responses Contribute to the Course of COVID-19?

(A)Innate immune response (recognition, interferon, and inflammasome activation)

After entering target cells via the ACE2 receptor, SARS-CoV-2 is detected by pattern-recognition receptors that resemble Toll-like receptors 3, 7, 8, and 9, as well as by the viral-infection sensors RIG-I and MDA5 [14,32]. The recognition stimulates the type I interferon (IFN) response and activates IFN-dependent genes [14,33]. The release of danger-associated molecular patterns (DAMP) contribute to the activation of the NLRP3 inflammasome [34] and other inflammasome complexes. The NLRP3 inflammasome induces caspase-1-dependent cleavage and the release of the key proinflammatory cytokines interleukin-1ß (IL-ß) and IL-18, and this correlates with COVID-19 disease severity [14,35]. Furthermore, gasdermin-D-mediated pyroptotic cell death is triggered [14,36]. Due to cell death, lactate dehydrogenase (LDH) is produced. These data underline that inflammasome activation is an important feature of COVID-19 which is reflected by high LDH blood levels [14,37]. Similar to SARS-CoV and MERS-CoV, SARS-CoV-2 is able to inhibit type I IFN responses in infected cells [38,39,40,41]. This allows the virus to replicate itself and to induce stronger tissue damage, and therefore an overwhelming immune response (see Figure 2, severe SARS-CoV-2 disease). Hence, the immune system struggles to limit viral replication while managing dying and dead cells [14]. Immune cells begin to also flow into the lungs, where they produce high amounts of proinflammatory cytokines, a process that escalates the situation [14]. The imbalanced immune reaction caused by the impaired type I IFN response contributes significantly to the overall severity of acute COVID-19 [13,41,42,43] by inducing a prolonged and ineffective innate immune response (Figure 2). This is also emphasized by new results coming from the COVID Human Genetic Effort, which found that inborn errors in the type I IFN pathway [44], or the presence of neutralizing autoantibodies to type I IFNs [11,45], were over-represented among individuals who developed life-threatening COVID-19. If these imbalanced or impaired innate responses also contribute to the development of MIS-C and long COVID remains to be determined [14]. In any case, an overwhelming inflammasome activation is an important component of severe COVID-19 [37]. This pathway also triggers a coagulation cascade which explains coagulopathy and severe thrombotic events in patients with severe COVID-19 [36,46]. Figure 2 shows the different kinetics shown in an average and in a severe SARS-CoV-2 infection. During a simple SARS-CoV-2 infection, the innate immune response phase gains rapid control of the viral load and is replaced by an effective and adaptive immune response based on adequate antibodies and T cell control. During a severe infection, the balance between innate and adaptive immune responses is essentially disturbed. An ineffective innate immune response with a probably weak type I IFN response does not gain control over the virus, and therefore prolongs and overlaps with an ineffective adaptive immune response. An ineffective adaptive immune response is characterized by a very weak and delayed adaptive T cell response. This situation allows the virus to expand, which usually leads to an overwhelming systemic inflammatory reaction.

(B)Adaptive immune response (Antibody production, seroconversion, T cell memory development)

The adaptive immune responses induced by SARS-CoV-2 infection largely follow the expected patterns observed in other comparable viral infections. Thus, more than 90% of infected individuals seroconvert a few weeks after initial infection [47,48], and anti-spike IgG antibodies are associated with protection from reinfection. T cell responses to the SARS-CoV-2 spike protein parallel B cell responses to the similar protein in nearly all COVID-19 cases [49] (Figure 2). Additionally, unexposed people occasionally show T cell reactivity to SARS-CoV-2 due to cross-reactive immunity to common cold coronaviruses [49]. The so-called antibody-dependent enhancement (ADE) has been supposed as another possible mechanism of severe COVID-19. These antibodies can facilitate Fc-receptor-mediated endocytosis of the virus and enhance viral replication resulting in massive hyperinflammation. ADE has been proved in dengue [50] and MERS [51] but clear evidence for its involvement in severe SARS-CoV-2 infections as well as SARS-CoV-2 second infections is lacking so far (https://doi.org/10.1016/S1473-3099(20)30783-0, accessed on 2 May 2021). Furthermore, to what extent this putative mechanism may also influence the balance between innate and adaptive immune responses in severe COVID-19 as shown in Figure 2 is not known so far.

### 2.4. How Strongly Does the Immune Response to SARS-CoV-2 Differ between Individuals?

Immune responses can vary markedly. Some people generate a very effective immune response. They cannot be infected again, and they will not give the virus to anyone else (so called ‘sterilizing’ immunity). Other people produce antibodies and are protected from disease, but not from future infection by SARS-CoV-2 [28]. These people still pass the virus. The number of antibodies produced after infection, the quality of those antibodies (how good they are at preventing infection) and the number and quality of the T cell response generated vary markedly between individuals. It is not known in detail so far for how long and how strong this immunity is over a longer period of time. With the rapidly growing proportion of the world’s population having been infected with SARS-CoV2, the degree to which this group is protected from reinfection is increasingly of interest. The waning of neutralizing antibody responses over the first year after infection makes reinfections more possible [53]. However, B cell and T cell memory responses induced by primary infection suggest that reinfection severity, and potentially transmission, may be mitigated over the longer term [53]. The potential of higher levels of neutralizing antibodies to be induced by vaccination suggests that reinfection could be further reduced by vaccination of those who have previously been infected [53]. Undoubtedly, a better understanding of all those factors, which influence immunity, is urgently needed to help achieve long-term immune control of the SARS-CoV-2 pandemic.

(A)The influence of age and lifestyle

Age is an important factor. Usually, the risk for severe COVID-19 increases abruptly above age 70 [54,55,56]. Nevertheless, this may not be true for all elderly people. Some studies reported that antibody responses to infection do not vary with age in adults [57] and were even higher in older patients [58,59]. This was recently confirmed in 217 participants obtained from the Austrian Ischgl cohort by a 7–8 months (follow up) after infection [60]. Lifestyle factors associating with increased low-grade inflammation may explain these discrepancies. Preexisting obesity, hypertension, chronic obstructive pulmonary disease, cardiovascular disease (CVD) and smoking associate stronger with severe COVID-19 courses [61]. Notably, smoking induces, besides other negative effects, the expression of angiotensin-converting enzyme 2 (ACE2), which allows SARS-CoV-2 to enter cells [62]. Furthermore, an increased neutrophil to lymphocyte ratio (NLR) is a surrogate marker for systemic inflammation and is associated with poor prognosis in COVID-19 [63]. The NLR increases with the degree of obesity especially in context with metabolic syndrome and type 2 diabetes [64]. Older individuals with such conditions show two characteristics of severe COVID-19, such as failure in developing sufficient antiviral immune responses and a tendency to develop uncontrolled exacerbating responses to infections resulting in hyperinflammation and acute respiratory distress syndrome [14]. Older individuals also have weaker type I IFN responses, which further aggravate situations [65]. Also, additional “inflammaging” markers like NLPR3 activation [37], IL-6, IL-12 and IL-1ß secretion [66], and danger-related molecular patterns, including high mobility group box 1 (HMGB1) [67] have shown to be predictive for a severe COVID-19 course in elderly. The majority of young people have mild COVID-19 disease [68]. This is interesting because, although newborns and young children produce lower amounts of type I IFN upon stimulation through the virus and show a reduced breadth of antibody responses to SARS-CoV-2 proteins, they experience mild COVID-19 [58,69]. Possible explanations may be given to this by protective action taken by cross-reactive antibodies to common-cold coronaviruses and constitutive differences of the immune system [70]. Furthermore, the immune system of young children is more accustomed to face novel challenges, while the elderly mainly rely on memory driven immune responses [14]. This may explain the more benign clinical courses in the young.

(B)The influence of gender

Men have a much greater risk of severe COVID-19 courses but women more frequently develop long COVID [71]. It is interesting that, in this context, women elicit stronger type I IFN responses after stimulation with TLR7 ligands [72], that they develop stronger vaccine responses and that they have better survival rates for a number of acute infections than men [73]. These sex differences are not age dependent and, thus, can also be seen before puberty between boys and girls, which indicates rather genetic differences. Interestingly, TLR7, a common virus sensor, is expressed on the X chromosome, suggesting a possible gene-dosage effect between the sexes [74]. Furthermore, neutralizing autoantibodies to type I IFN are much more frequently found in male patients with COVID-19 [14]. Future research is warranted to elucidate the reason for this phenomenon. The thymus involutes more rapidly in boys than in girls which may explain a weaker T cell control in men [75]. Otherwise, MIS-C is quite evenly distributed between boys and girls [76]. It is also important to which extent social factors and differing exposure play a part in sex differences [14].

(C)The influence of immune deficiency

People with immunodeficiency, autoimmune diseases, or those who take immunosuppressive medication respond less to SARS-CoV-2 than young healthy people [77]. However, the data are also contradictory for this. One systematic review found no statistically significant increased risk of severe COVID-19 in immunosuppressed persons [78] whereas in cases with solid-organ transplants and cancer patients an increased risk was seen [79]. Especially, cancer patients treated with checkpoint inhibitors may be at a high risk to develop severe COVID-19 [80]. On the other hand, it is also possible that the host immune response against SARS-CoV-2 can cause an anticancer effect in certain constellations. Challenor et al. [81] reported one case with classical Hodgkin lymphoma with stage III disease that went into remission without corticosteroids or immunochemotherapy. Thus, the SARS-CoV-2 triggered immune activation may mediate an anti-tumor immune response as seen in other infections in the context of high-grade non-Hodgkin lymphoma [82]. Future research is needed to clarify the outcome of cancer patients who overcome COVID-19 disease. Particularly tumor patients with good checkpoint inhibitor effects that successfully overcome COVID-19 disease should be subject to research [83]. Furthermore, the effects of immunosuppressive medications, especially methotrexate and rituximab, on decreasing serological responses must be determined in future studies. Concerning successful vaccination against SARS-CoV-2 in rheumatic diseases, methotrexate may be held for up to 2 weeks after the vaccination, and rituximab a few weeks after the vaccination until further clinical trials can answer this question [84]

### 2.5. What Are the Characteristics of an Effective Immune Response after Confirmed Infection?

In general, after viral infections, antibody concentrations decline over time depending on the type of virus causing the infection. For example, Respiratory Syncytial Virus (RSV) elicits a very short-lived immunity, whereas immunity to measles stays lifelong [85]. Infection with HIV is fatal because damage of CD4^+^ T cells leads to a breakdown of the immune reactivity. 

If specific antibodies to pathogens are lost, immunity can be reactivated. By new contact with the virus, memory T cells induce new antibody-secreting cells from the memory B cell population. This can occur naturally, by vaccination, or by the booster vaccination, respectively. This outlines that measuring antibodies is a limited indicator of immunity, and that the determination of memory lymphocytes of different types may be a better one. After undergoing COVID-19, around 90% of recovered patients have detectable anti-SARS-CoV-2 antibodies for several weeks or months after their infection [47]. If, and how long, this is effective enough to prevent a reinfection remains to be clarified. Studies in healthcare workers showed that presence of SARS-CoV2-specific antibodies offer approximately 95% protection against COVID-19 symptoms and about 75% protection against being infected [28]. Presence of SARS-CoV-2 responsive CD4^+^ T cells is definitely protective, as found in convalescent COVID-19 patients [86]. Respectively, it was shown that about 7 months post-onset of COVID-19 symptoms, humoral responses, including titers of the spike receptor-binding domain IgG and neutralizing antibody start to decrease significantly compared with those at the first clinic visit. By contrast, the proportions of spike-specific CD4^+^ T cells, but not CD8+ T cells remained persistently higher after recovery than those in healthy controls [86]. Thus, the SARS-CoV2-specific CD4^+^ T-cell immune responses persists, while the humoral immune response decays [86]. Further studies are needed to evaluate whether these T cells are sufficient to protect patients from reinfection [87,88]. In a recent review, we discussed a modulating/suppressing role of ACE2 on the function of B cells which may be relevant for long term immune competence after infection with SARS-CoV-2 in patients with cardiovascular diseases (CVD) [89]. This B cell response modulating/suppressing role of angiotensin 2 remains to be elucidated for a contributive role to weaker immune responses to COVID-19 infections found in patients with active CVD [89].

### 2.6. Measuring Anti-SARS-CoV-2 Immune Cells and Antibody Responses

Immune cells come in various forms, most important are the T cells. Compared with antibody tests, detection of antiviral T cells is time-consuming, technically challenging and not easily automated or scaled-up for large numbers of samples. As a consequence, these tests are limited to specialist labs and in-depth studies of immunity. Later post-infection, “memory” immune cells—either T-memory or B-memory cells, persist and can be measured by even more complex tests. Currently, the best marker is the level of neutralizing antibody in the blood. This component of the antibody repertoire avoids viral entry into cells. While a decline in neutralizing antibody is seen [90], it can persist in individuals for at least 8 months, and possibly longer, after infection [29]. B and T lymphocytes recognizing SARS-CoV-2 and T cells capable of mounting robust responses have been detected after 8 months [29] and longer [88]. Ongoing research is required to fully understand how long immunity lasts after infection. T cell-mediated response is evaluated by stimulating immune cells (whole blood or isolated PBMCs) with synthetic peptides based on SARS-CoV-2 antigenic proteins and evaluating IFNγ release, measured by enzyme-linked immunosorbent spot or the ELISA test. Interferon-γ is a cytokine playing a fundamental role in the clearance of viral infection. However, it has been shown that the magnitude of the IFNγ-secreting T-cell response may not be sufficient to ensure effective immune protection. The evaluation of Th1 cytokine polyfunctionality response, characterized by co-production of IFNγ, TNF and IL-2, seems to be of major importance for the evaluation of viral-specific T cell responses [91]. Notably, the composition and degree of purity of the antigenic peptides used may make the results obtained in the different studies difficult to compare.The analysis of T cell response, supplemented by serological data, gives a comprehensive view of the individual immune response to SARS-CoV2. This analysis is useful for determining previous exposure to SARS-CoV-2 and is fundamental to evaluate protection against reinfection or the response to vaccine in elderly or immunocompromised patients. It has been demonstrated the B-cell depletion following rituximab treatment impairs serological responses, while T-cell responses were preserved [92]. Further, cellular immune responses were not attenuated in patients receiving methotrexate or targeted biologics compared with controls [93]. Nevertheless, the efficacy and the longevity of cell-mediated immune responses against SARS-CoV-2 in immunocompromised patients remains to be clarified.

### 2.7. Does Viral Load Influence Levels of Immunity?

Severely ill patients generally have higher viral loads [94], and some weeks later, higher levels of antibodies than patients with mild COVID-19 [95]. In one study, in the acute phase of illness (2–3 weeks) higher viral loads paralleled with higher IgM antibody levels [96]. IgM antibodies are rapid responders to the infection but are not that good at fighting the virus and do not last very long. In contrast, the highly protective IgG antibody levels did not reflect the viral loads or were even negatively correlated with them [97].

### 2.8. Reinfection with SARS-CoV-2—When Does It Occur, and How Can It Be Verified?

Reinfection is not very frequent [98]. A study in Mexico found only 258 instances of reinfection in a cohort of 100,432 participants (2.6 per 1000) [99]. Reinfection depends upon both, the level of immunity a person gains from the first infection, and the level of likely exposure due to social and working environments [100]. To exclude persistence of viral RNA, reinfection is proven only when both the first and the second infections have been sequenced and shown to be different variants of the virus [100]. A less rigorous way of looking for reinfection is to document a positive PCR test at a set time interval after the last known PCR or antibody test [98]. Experience with the SARS-CoV-2 vaccination programmes showed that protective immunity gained by natural infection with SARS-CoV-2 is poor compared to the much higher levels of virus-neutralizing antibodies and T cells induced by the vaccines [101,102]. An emerging problem is the development of new variants of SARS-CoV-2 with the ability to escape natural and vaccine-induced immunity [103,104]. This fact prompts the need for updated vaccines [105], and confirms that a global vaccination program with high efficacy vaccines is one of the most important challenges of the next years.

### 2.9. Could There Be Any Cross-Immunity with the Cellular Response from Other Coronaviruses?

Studies have now reported the presence of T cells [106] that can recognize both SARS-CoV-2 and other coronaviruses causing common colds. One such study indicated that over 80% of uninfected individuals had T cells capable of recognizing SARS-CoV-2 [107]. It remains to be clarified if these cross-reactive responses are protective. A recent study found that health care workers who had such T cells were less likely to catch COVID-19 than their colleagues who did not have such cells (https://doi.org/10.1101/2020.11.02.20222778, accessed on 2 May 2021). Similarly, antibodies that can neutralize SARS-CoV-2 have been found in a small proportion of blood samples taken from people prior to the SARS-CoV-2 pandemic [108]. These are thought to be cross-reactive with common cold coronaviruses and were found to be more prevalent in children.

### 2.10. How Does Pregnancy Influence Immunity?

The maternal, neonatal, and perinatal outcomes of COVID-19 patients infected in late pregnancy were recently reported to be favorable [109]. Transplacental transmission of SARS-CoV-2 to the fetus can occur, but the immediate and long-term effects on the newborn infant remain unclear. Therefore, women who are pregnant or planning a pregnancy should be managed according to current clinical guidelines with timely vaccination to prevent infection with SARS-CoV-2 [110]. Concerning vaccination, as shown by the adverse effect profile and short-term obstetric and neonatal outcomes, no safety signals appeared among pregnant women who were vaccinated by BNT162b2vaccine at all stages of pregnancy. The vaccine is effective in generating humoral immune response in pregnant women, although IgG levels were lower than observed in non-pregnant women [111]. In a review of 38 Chinese pregnant women suffering from COVID-19, no fatality occurred to mothers and their 39 children. Comorbid maternal conditions like preeclampsia, pregnancy-induced hypertension, [109] uterine scarring, gestational diabetes, and uterine atony did not appear to be risk factors for intrauterine transmission of SARS-CoV-2 to the fetus [112]. SARS-CoV-2 does not belong to the congenitally transmitted TORCH agents (acronym for Toxoplasma, other, rubella, cytomegalovirus, herpes) that also include Zika virus and Ebola virus. In a recent meta-analysis from China including 324 pregnant woman, it was criticized that, despite the increasing number of published studies on COVID-19 in pregnancy, there is insufficient good-quality data available to draw solid conclusions with regard to the severity of the disease or specific complications of COVID-19 in pregnant women. This includes vertical transmission, perinatal and neonatal complications [113]. Altogether, uncertainty remains. Beside the direct impacts of the disease, a plethora of indirect consequences of the pandemic may adversely affect maternal health, including reduced access to reproductive health services, increased mental health strain, and increased socioeconomic deprivation [114]. More data is necessary on the effect of COVID-19 vaccination, in terms of maternal and fetal outcomes as well as vaccine related symptoms in high risk women with obesity and diabetes during pregnancy and breastfeeding [115]. In a recent cohort study, it was shown that maternal IgG antibodies to SARS-CoV-2 are transferred across the placenta after asymptomatic as well as symptomatic infection during pregnancy [116]. Cord blood antibody concentrations correlated with maternal antibody concentrations and with the duration between onset of infection and delivery [116]. These findings demonstrate the potential for maternally derived SARS-CoV-2 specific antibodies to provide neonatal protection from COVID-19 [116].

On the one side, one ought to consider that up to two weeks counting from complete vaccination (2-dose series with an interval of 21 or 28 days or more between doses with current COVID-19 vaccines) are necessary for higher efficacy of the vaccine. On the other side, it is known that transplacental transfer begins around 17 weeks of gestation increasing exponentially as gestation advances and the placenta grows. Accordingly, maternal vaccination starting in the early second trimester of gestation might be optimal to achieve the highest levels of antibodies in the newborn [117]. While a serologic correlation of protection against SARS-CoV-2 infection and symptomatic or severe disease is currently unknown, higher antibody levels might result in a better chance for protection of the newborn during a period of special vulnerability [117].

### 2.11. How Does Severe COVID-19 Manifest in Children and Young?

Although infection with SARS-CoV-2 is usually mild in children, some children develop a severe inflammatory disease later that can have manifestations similar to toxic shock syndrome or Kawasaki disease [118,119]. This syndrome has been defined by the US Centers for Disease Control and Prevention as the so-called multi-systemic inflammatory syndrome in children (MIS-C). Even though the prevalence is unknown, so far more than 600 cases have been reported. Multisystem inflammatory syndrome in children appears to be more common in Black and Hispanic children [119]. The reason why children of these ethnic groups are more affected remains unclear. It is suspected that genetic factors (HLA-types?) which cause a stronger disposability to certain autoimmune-diseases later in life [120], and/or social factors [121] may contribute to this burden. A recent European study compared blood immune cells, cytokines, and autoantibodies in healthy children, children with Kawasaki disease enrolled prior to COVID-19, children infected with SARS-CoV-2, and children presenting with MIS-C [122]. It demonstrated that the inflammatory response in MIS-C differs from the cytokine storm of severe acute COVID-19 shares several features with Kawasaki disease, but also differs from this condition with respect to T cell subsets, interleukin-17A, and biomarkers associated with arterial damage [122]. An autoantibody profiling performed in this study suggested multiple autoantibodies that could be involved in the pathogenesis of MIS-C [122]. MISC-C typically occurs a few weeks after acute infection. The putative etiology is a dysregulated inflammatory response to SARS-CoV2. Persistent fever and gastrointestinal symptoms are observed. Cardiac manifestations are common, including ventricular dysfunction, coronary artery dilation and aneurysms, arrhythmia, and conduction abnormalities. Severe cases can present themselves in the forms of vasodilatory or cardiogenic shock requiring fluid resuscitation, inotropic support, and, in the most severe cases, mechanical ventilation and even extracorporeal membrane oxygenation. Empirical treatments include intravenous immunoglobulin, steroids, and other immunomodulatory agents. Medium- and long-term sequels, particularly cardiovascular complications, are not yet known [119]. Although most children require intensive care and immunomodulatory therapies mortality rates remain low [123].

## 3. Questions Addressing the Vaccination against SARS-CoV-2

### 3.1. What Form of Immunity and Protection Does the Vaccination Provide?

So far, five vaccines (BioNTech/Pfizer, Moderna, Oxford/AstraZeneca, Novovax, and Johnson & Johnson) strongly reduced the number of people becoming sick from COVID-19. At 1 August 2021 (source: Coronavirus (COVID-19) Vaccinations—Statistics and Research—Our World in Data) 4.1 billion doses were vaccinated at least 1 time world-wide. The vaccines clearly prevent both mild/ moderate and severe COVID-19 symptoms [124,125,126]. It is unknown if the vaccines can prevent infection with the virus, or whether it can counter becoming infectious for other people. Many teams are currently busy with assessing the incidence of asymptomatic infection and viral shedding after vaccination [126]. Perhaps, a longer interval between the vaccination doses promotes a better immune response as shown with the Pfizer-Biontech mRNA vaccine https://www.medrxiv.org/content/10.1101/2020.11.02.20222778v2, accessed on 16 July 2021. New data suggest that heterologous vaccinations with vector as first and mRNA vaccine as second shot can provoke a better immune response to SARS-CoV-2 than homologous vaccinations with either Vaxzevria, AstraZeneca or mRNA vaccines. Thus, in light of changing recommendations regarding use of Vaxzevria, several countries are now advising individuals previously primed with this vaccine to receive an alternative vaccine as their second dose, most commonly mRNA vaccines such as the BNT162b2 (BNT) COVID-19 vaccine (Comirnaty, Pfizer-BioNTech), administered in a heterologous prime-boost schedule (https://www.has-sante.fr/jcms/p_3260335/en/covid-19-quelle-strategie-vaccinale-pour-les-moins-de-55-ans-ayant-deja-recu-une-dose-d-astrazeneca, accessed on 9 April 2021). Although heterologous vaccine schedules induced greater systemic reactogenicity following the boost dose than their homologous counterparts [127,128], they may have advantages in form of stronger efficacy against multiple SARS-CoV-2 variants, including the worldwide arising Delta variant https://www.medrxiv.org/content/10.1101/2021.07.13.21260417v1, accessed on 16 July 2021; https://www.medrxiv.org/content/10.1101/2021.05.19.21257334v2 [129], accessed on 2 June 2021.

Recently, the appearance and spread of new SARS-CoV-2 variants have produced enormous concern due to their possible implication in the improved transmissibility of the virus, their consequences in the individual evolution of the infection, as well as in the possible escape from the immunity generated by the current vaccines [130]. The variants that attracted most attention are those of public health concern, including B.1.1.7 (UK, Alpha), B.1.351 (South African, Beta) and P.1 (Brazilian, Gamma). This list is extended by the variants of interest that emerged and are expanding in certain countries but are solely found sporadically in others, such as B.1.427 and B.1.429 (Californians) or B.1.617 (Indian, Delta) [130] and C.37 (Lambda) [131]. Whole genome sequencing or strategies specifically targeting the spike gene are used for characterization and detection. So far, it is not excluded that revaccination against new variants may be necessary in the future [130]. To date, a low or insignificant impact on vaccine efficacy against Alpha and Delta variants has been reported [131]. Such an impact on vaccine efficacy for Beta, Delta, Gamma, and Lambda variants may be even greater compared to the Alpha variant [131]. More comprehensive analyses are needed to assess the real impact on vaccine efficacy brought about by SARS-CoV-2 variants [131].

### 3.2. Experiences from a Large Vaccination Campaign in the Graz Medical Hospital in Austria

The University Hospital of Graz organized a vaccine campaign for their employees as well as for their critically ill patients (patients with an oncologic disease, transplant-patients, rheumatologic patients, etc.). Between January and June 2021 in total 14,970 vaccinations for employees and 7600 vaccinations for patients were done. Patients received explicitly RNA based vaccines (100%), employees received either a vector (75%) or an mRNA vaccine (25%). The choice of mRNA or vector vaccine was triggered by availability. In general, adverse effects were seen more often for the vector vaccine dominantly after the first shot. Around 30% of the health care workers suffered for one up to three days of well described adverse reactions (fatigue, fever, chills, head ache, muscle pain, etc.). After the second shot the rate of adverse events decreased to approximately 15%. Allergic reactions were observed rarely; i.e., only one for the vector and one for the mRNA vaccine. In one case, the allergic reaction led to hospitalization for one day. In a few cases it was observed that employees as well as patients who had been concerned about the vaccination reported pseudo allergic symptoms. A limitation of these data is given by the fact that the rate of adverse events was monitored only for up to two weeks after the shots. Nevertheless, the experience with all three vaccines (Astra Zeneca, Biontech Pfizer and Moderna) was generally quite good.

### 3.3. How Will Pregnant Women React to Vaccination?

Pregnant individuals have traditionally been excluded from clinical trials of new medications and vaccines because of concerns over possible effects on the fetus. In summary, pregnant individuals and their obstetricians will need to use the limited available data to weigh the benefits and risks of COVID-19 vaccine during pregnancy, taking into account the patient’s specific risk of SARS-CoV-2 exposure [132].

### 3.4. What Do We Know about the Clotting Disturbances after SARS-CoV-2 Vaccination?

It is now well established that COVID-19 also affects the vascular [133,134,135,136,137,138] and the clotting system [139,140,141,142,143,144]. Thus, it is not surprising that SARS-CoV-2 vaccinations were brought into connection with clotting disturbances including serious thrombotic thrombocytopenia. Two research teams reported detailed observations of this pathological reaction after vaccination with the AstraZeneca vector vaccine [145,146]. They suppose that the development of serious blood clots is an immune response that resembles a rare reaction to the drug heparin, called heparin induced thrombocytopenia (HIT). The researchers named this phenomenon vaccine induced immune thrombotic thrombocytopenia (VITT). Among more than 20 million people who have been vaccinated with the AstraZeneca vaccine in the UK so far, only 79 cases of rare blood clots with low platelets have been reported. Unfortunately, 19 deaths were also recorded. Relating to these events, the European Medicines Agency and the UK’s Medicines and Healthcare Regulatory Agency have concluded that unusual blood clots with low blood platelets are a possible and rare side effect of the AstraZeneca vaccine although a causal relation remains unclear [147]. The EMA’s Pharmacovigilance Risk Assessment Committee also continue to investigate three cases of unusual blood clots with low blood platelets during the vaccine rollout of the Johnson &Johnson Janssen vaccine in the USA. The first paper, published in the New England Journal of Medicine [146], described 11 patients in Austria and Germany, nine of them women, with a median age of 36 years that had clotting and low platelets between five to 16 days after vaccination. Nine patients had cerebral venous thrombosis, three had splanchnic vein thrombosis, three had pulmonary embolism, and four had other thromboses. Six of the patients died. All 11 patients, as well as another 17 for whom the researchers had blood samples, were positive for antibodies against platelet factor 4 (PF4). These antibodies occur also in people who develop HIT. Nevertheless, none of the vaccine induced incidents had heparin before their symptoms started. Whether these antibodies are autoantibodies against PF4 induced by strong inflammatory stimulus of the vaccination or antibodies induced by the vaccine that cross-react with PF4 and platelets requires further investigations. One possible trigger of PF4 reactive antibodies could be free DNA in the vector vaccine of AstraZeneca. In a second paper, also published in the New England Journal of Medicine [145], researchers in Norway describe a similar pattern in five healthcare workers between 32 and 54 years old. All had high levels of PF4 antibodies and no previous exposure to heparin. Four of the patients had major cerebral hemorrhage and three died. Thus, enzyme linked immunosorbent assay testing for PF4-heparin antibodies in patients who have unexpected symptoms after vaccination is strongly recommended. Treatment with intravenous immune globulin and non-heparin blood thinners is helpful. On 7 April 2021, the UK’s Expert Hematology Panel published guidance for the diagnosis and management of vaccine induced thrombosis and thrombocytopenia which they say is rare but can affect patients of all ages and both sexes. They recommended urgent use of intravenous immunoglobulin, avoiding platelet transfusions, and anticoagulating with non-heparin based therapies. The Medicines and Healthcare products Regulatory Agency (MHRA) is recommending that anyone developing the following symptoms after vaccination should seek prompt medical advice: shortness of breath, chest or persistent abdominal pain, leg swelling, blurred vision, confusion of seizures, unexplained pin prick rash, or bruising beyond the injection site.

### 3.5. Own Experience with Cerebral Sinus Venous Thrombosis after SARS-CoV-2 Vaccination

In general, thrombosis of the dural sinus and/or cerebral veins (CVT) accounts for less than 1% of all strokes and predominantly affects young female adults. Four out of five CVT patients have at least one classical risk factor for venous thrombotic events, such as pregnancy, puerperium, and exposure to drugs (e.g., oral contraceptives), hereditary or acquired thrombophilia or smoking [148].

Thrombocytopenia-associated CVT is a very rare clinical condition, and has mainly been reported in hemato-oncological disorders or autoimmune disease, including heparin induced immune thrombopenia (HIT) [149]. Nevertheless, some cases of CVT and severe thrombocytopenia have recently been observed after vaccination against SARS-CoV-2 using the AstraZeneca (ChAdOx1 nCoV-19) or the Johnson & Johnson/Janssen (Ad26.COV2.S) vaccines [150]. This phenomenon was termed VITT. Although an immune-mediated PF4-antibody-related mechanism similar to HIT seems likely, the exact patho-mechanism of VITT still needs to be elucidated [145,146,150].

In the tertiary care university hospital of Graz, Austria, experience with two young women with VITT-associated CVT was recently gained. Both patients presented a severe, persisting headache one to two weeks after receiving their first ChAdOx1nCov-19 vaccination. Laboratory examinations yielded thrombocytopenia (<40 × 10^9^/L), fibrinogen depletion (<150 mg/dl) and significantly elevated D-dimer levels (>10 mg/L). Initial PF4-antibody tests (HemosIL HIT IgG) were negative in both patients, which were subsequently tested positive using a confirmatory ELISA assay. In accordance with previously published reports [151], both patients were treated with a combination of high-dose intravenous immunoglobulins (IVIG), corticosteroids and anticoagulation with the thrombin-inhibitor argatroban. After recovery of platelets and clinical stabilization, argatroban was replaced by the oral anticoagulant dabigatran. Both patients fully recovered (Gattringer et al., 2021 in press) [152].

Since high VITT-associated mortality rates of up to 40% have been reported, early recognition and prompt initiation of targeted treatment regimens based on interdisciplinary expertise, including vascular neurology, hematology and vascular medicine, is crucial in these cases [153]. Although there is limited information about optimal treatment, IVIG (1 g/kg body weight for two days) and high-dose steroids have been shown to improve platelet count within a few days. Additional anticoagulation with non-heparin products (e.g., thrombin-inhibitors) may be used to prevent progressive thrombosis and consecutive brain damage despite low platelet count and fibrinogen depletion. Platelet transfusion, fibrinogen supplementation and coagulation factor concentrates should be avoided [153]. PF4-antibody testing is recommended, but might not be available 24/7, could be false negative, and should therefore not delay treatment [154].

### 3.6. Which Individuals Will Not Generate an Effective Immune Response following Vaccination?

Older and immunocompromised people might not make a comparable effective immune response as young, healthy individuals. However, current data exists that the vaccines generate similar immune responses in older and younger people [124,155,156]. There are limited efficacy data specifically for older individuals at present, but this will continue to be evaluated [55]. None of the Phase III trials enrolled immunocompromised individuals [55]. Nevertheless, vaccination is still recommended as immunocompromised individuals may be at higher risk of severe SARS-CoV-2 infection courses. People with immunosuppressive medications will need special considerations. As previously mentioned, since those patients were excluded from vaccine trials, the efficacy of the vaccine on them still needs to be established. The effect of immunosuppressive medications, especially methotrexate and rituximab on a SARS-CoV-2 vaccine response, is yet to be determined and will need evaluation, especially given its’ influence on decreasing serological responses to other vaccines. Specific planning to vaccinate the immunocompromised patients to ensure maximum possible seroprotection will be necessary. Considerations can be given to holding methotrexate for 2 weeks after the vaccination, and scheduling rituximab a few weeks after the vaccination until further clinical trials can answer this question [84].

### 3.7. How Long Will Vaccination-Induced Immunity Last?

This is not yet known in detail, but so far, published data from the disease itself suggest that immunity will last around one year [29,53]. Comparable to influenza, revaccinating will be necessary every year like the vaccination programs for flu. Most mRNA-1273 vaccinated individuals maintained binding and functional antibodies against SARS-CoV-2 variants for 6 months [157]. Current vaccines may become less effective over time caused by mutant virus strains. Thus, periodic modifications comparable to the flu vaccines must be performed.

### 3.8. What Do We Need to Achieve Herd Immunity?

Herd immunity happens when transmission rates of the virus within a population markedly decline due to a high proportion of people already being immune [158]. If sufficient people in the population are immune, the virus remains at low or undetectable levels, thus protecting anyone who is not yet vaccinated (e.g., infants), those unable to make a good immune response themselves (e.g., people who are frail, very elderly or immunocompromised), or those who are allergic to components of the vaccine. First and foremost, for a vaccine to confer herd immunity, it has to either stop or substantially reduce transmission [158]. If the vaccine prevents symptoms but has small effect on infections, it cannot confer herd immunity. This is the result of vaccinated people still getting infected and continuing to transmit the virus without getting the COVID-19 disease themselves. The proportion of the population who has to be immune, or otherwise not susceptible, in order to stop transmission depends on how infectious that pathogen is, how long someone remains infectious (few days, months or years?) and whether people know they are infected and infectious [158]. Overall, the number of people required to have immunity is dependent on the R0 value for the virus. In the case of SARS-CoV-2 and vaccine efficacies of ~95% it is estimated that 63–75% of people need to be immune to provide herd immunity [159,160]. Hence, a high acceptance of vaccines, or a widespread natural immunity following infection is required to achieve this [161]. Unfortunately, COVID-19 delta infections currently threaten vaccine strategies to achieve herd immunity [162]. Present results show that countries are far from the desired herd immunity threshold to safely slow down or stop the COVID-19 epidemic [163,164]. The concept of herd immunity during COVID-19 is constantly changing [165]. The World Health Organization’s prevailing focus is on vaccination. However, moral conservatism that tend to not cooperate in the rollouts present a problem [165].

### 3.9. Is There a Difference in the Herd Immunity Acquired from Natural Versus Vacine-Induced Immunity?

This remains unknown. However, predictions state that vaccine-induced immunity will be much more durable [165]. Most importantly, to achieve herd immunity, immune responses should reduce infectivity and transmission, and this should have a long-lasting effect [166].

### 3.10. Can the Vaccine Stop “Long-COVID”?

The vaccines that are currently approved have been shown to effectively reduce the chance of getting COVID-19. Hence, it is very probable that they will also reduce the risk of any long-term effects of COVID-19 [167]. Some patients with Long-COVID are concerned about being vaccinated, either because they feel they made an inadequate immune response to the natural infection so would not benefit, or because they suspect they may have made an excessive or dysregulated response to natural infection that may be exacerbated by further immune stimulation. Any data confirming or refuting these suspicions are not yet available, but these data should emerge over the next few months as more and more people with long COVID are vaccinated. In the meantime, the assumption is that, like others, people with long COVID would benefit from vaccination to reduce their risk of further infection [167]. As discussed recently, vaccination may tackle residual virus, or, if the mechanism is autoimmune, “reset” the immune system [167,168], https://www.yalemedicine.org/news/vaccines-long-covid (accessed on 12 April 2021). A small study, which has yet to be published in a peer-reviewed journal found that the AstraZeneca and Pfizer-BioNTech vaccines were associated with overall improvements in symptoms of Long-COVID. There was no evidence of declines in quality of life or mental well-being https://www.medrxiv.org/content/10.1101/2021.05.19.21257334v2 (accessed on 12 April 2021), https://doi.org/10.1101/2021.05.19.21257334 (accessed on 2 June 2021). SARS-CoV-2 infection leads also to increased numbers of double negative B memory cells [168], which are described as a dysfunctional B cell subset. This effect was reversed by SARS-CoV-2 vaccination, providing a potential mechanistic explanation for a vaccination-induced reduction in symptoms in patients with Long-COVID [168].

### 3.11. Does Changing the Interval between First and Second Dose Affect Immunity and How Should Convalescents Be Treated?

There is evidence from the AstraZeneca/Oxford vaccine that a longer interval between vaccine doses improve immunity [125]. Recently, an interval of 12 weeks between the two required jabs was confirmed as optimal [169]. There is no such evidence from the Pfizer/BioNTech vaccine. For all currently available two shot based COVID-19 vaccines, it is critical that everyone receives both doses of the vaccine to maximize the immunity conferred. The requirement for boosting the response after priming with the first dose is uncertain in convalescents already primed by the natural infection. Evidence suggests that, after a single vaccine dose, convalescents develop antibody (total and neutralizing) levels similar to the ones measured in vaccinated people after the full two-dose course [170]. While concerns on the equivalent duration of such responses remain, optimizing vaccine delivery to convalescents seems effective and could accelerate achievement of herd immunity [170].

### 3.12. What Is the Difference between Naturally Acquired Immunity and Vaccine-Mediated Immunity?

The obvious advantage of a vaccine is that any side effects (e.g., fever, headache, muscle aches, arm pain), if present at all, are relatively mild and short-lived (usually resolved within 72 h). With very rare exceptions, these are insignificant in comparison to the much more serious consequences for individuals of any age who develop COVID-19 disease following natural infection [171,172]. Another difference between vaccination and infection is that the dose of vaccine you will receive has been tested to ensure the development of a good immunity. In contrast to this, the amount of virus exposure during a natural infection is uncontrollable, either too little to induce an optimal immune response or too much, so that it causes a potentially life-threatening disease. The currently available SARS-CoV-2 vaccines are built to provoke an assessable immune response to a small number of virus components in a stabilized conformation that likely maximize immune responses [172]. Hence, the immune response induced by these vaccines is highly targeted to crucial components of the virus whereas the response to natural infection is broader, including responses to less critical or unimportant components of the virus. Natural infection may also stimulate the production of undirected autoantibodies and tissue damage due to uncontrolled inflammatory responses [173]. On the other hand, it is possible that a broader response provides better protection against an expanding range of virus variants while the immune response of the vaccination is too small [31]. Some common coronaviruses have the ability to subvert an effective immune response to SARS-CoV-2 [174]. So, in more severe cases of COVID-19 disease, the immune response may not be able to work optimally. Hence, it is very probable that vaccination is a much safer way to gain protection than through natural infection. A major drawback is given by the development of emerging virus variants. Thus, current vaccines must be continuously refined to stay protective. Although the vaccines are an effective critical tool, none of them are a 100% effective in the prevention of COVID-19 illness [103]. A small percentage of the fully immunized population will show vaccine breakthrough cases. In the USA, more than 75 million people have been fully vaccinated as of 13 April 2021 since 14 December 2020 [103]. During the same duration, 5814 vaccine breakthrough cases were reported [103]. The vaccine in these cases also remains the most important strategy to prevent severe illness and death [175].

### 3.13. How Does Infection-Acquired Immunity Interact with Vaccination?

In general, infection-induced immunity and vaccine-induced immunity act in a complementary fashion, https://doi.org/10.1101/2021.01.27.21250567, accessed on 31 January 2021. It is difficult to estimate the levels of cellular immunity gained from a natural infection. Thus, to the best of our knowledge, people who underwent a manifest infection should also be vaccinated. If they present very high anti SARS-CoV-2 antibody levels one vaccination shot may possibly be sufficient enough to protect them [170]. In any case, the vaccinator must calculate the individual risk of such constellations.

### 3.14. What Are the Options If the Immune System Doesn’t Respond to the Vaccine?

Although unrealistic to achieve, herd immunity is the best way to protect the people who cannot be vaccinated due to suffering from immunodeficiency. Certain vaccines may be more effective for some people than others. This includes different age groups, immunodeficiency, autoimmune disease, and cancers. The immune systems of patients with common variable immune deficiency (CVID) are damaged to the point that they cannot respond to a vaccine. They can be protected by periodically receiving passive immunization with immunoglobulins or specific monoclonal antibodies. Hence, effective treatment options are urgently needed for severe COVID-19 patients. Successes were reported using dexamethasone [176] for those COVID-19 patients affected with severe lung disease. Ongoing research to better understand the underlying drivers of disease severity, for example, https://doi.org/10.1186/ISRCTN50189673 (accessed on 31 January 2021), Clinical Trials.gov number NCT04381936. ISARIC-4C (https://isaric4c.net/, accessed on 31 January 2021) and GenoMICC (https://www.genomicsengland.co.uk/covid-19/, accessed on 31 January 2021) will identify more potential drug targets, allowing new therapies to be developed and tested.

### 3.15. What Are Strong and Measurable Correlations between Protection and Disease?

It is crucial to identify which immune markers, such as immunoglobulins, memory (B, T) cells predict who is immune to COVID-19 and who is susceptible to either mild or severe disease. Determinants of protective immunity against SARS-CoV-2 infection require the development of well-standardized, reproducible high-throughput antibody assays which can pinpoint people who are at a particular risk. They may also provide information on how often booster vaccinations might be needed and help in developing improved vaccines and immunotherapies [177,178,179]. Nevertheless, there are significant concerns about only using the antibody response in coronavirus infections as a sole metric of protective immunity [180]. Antibody response is shorter-lived than virus-reactive T-cells. Strong antibody response correlates with more severe clinical disease while T-cell response associates with less severe disease [180]. Furthermore, antibody-dependent enhancement of pathology and clinical severity has been described [180]. Data from coronavirus infections in animals and humans emphasize the generation of a high-quality T cell response in protective immunity [180]. Progress in laboratory markers for SARS-CoV-2 has been made with identification of epitopes on CD4 and CD8 T-cells in convalescent blood [49,181,182], These are much less dominated by spike protein than in previous coronavirus infections [180]. Although most vaccine candidates are focusing on spike protein as antigen, natural infection by SARS-CoV-2 induces broad epitope coverage, cross-reactive with other betacoronviruses [180]. It is crucial to understand the relation between breadth, functionality and durability of T-cell responses and resulting protective immunity [180]. If protective immunity fades after vaccination or if new patterns of disease emerge after SARS-CoV-2 natural infection, data correlating clinical outcomes with laboratory markers of cell-mediated immunity, not only with antibody response, may prove critically valuable.

New SARS-CoV-2 virus variants must be recognized as early as possible through global communication networks. The recognition of them should immediately lead to updated vaccine designs. Although they may have a stronger reactogenicity [127,128], heterologous vaccinations may be a stronger protection because they act complementary in stimulating the cellular and humoral immune response, and memory functions [129].

## 4. Conclusions

Although we have learned much about Sars-CoV-2 and COVID-19 since its detection in late 2019, many questions remain without satisfactory answers at the moment. Undoubtedly, this virus has come to stay. While it looks as though the pandemic is under control in early summer 2021, the development of the situation and what will be the case in autumn or winter 2021, and afterwards, remain a mystery. Currently (early August 2021), e.g., infection rates in the USA, Great Britain, Russia, Pakistan, and in several countries of the southern hemisphere warn us about new attacks of modified virus versions, notably the Delta variant. The worldwide economic constraint, urging populations to return to “normal” life, may also pave the way for further pandemic waves in late 2021. On the other hand, it is also possible that COVID-19 may wane completely defeated by consequent area-wide vaccination programs, as well as other factors not understood in depth so far. Apart from these uncertain prospects, the fight against new pathogen-induced diseases remains, and will surely become more difficult in the future. The increasing number of human beings and the intense exchange frequency by travelling, by international air traffic in particular, will contribute to its burdens. Our lessons learned from COVID-19 may hopefully help to improve our biological armament. Returning to the focus of this review, our best weapon against SARS-CoV-2 is a prompt, well-balanced, and effective, but not overwhelming, immune response. Concerning vaccination targets, so far, we have concentrated on the spike protein. This was, on the one hand very successful, but, on the other hand, it also implements some dangers. Firstly, the virus may effectively modify the building plan for the S protein by escape mutations. Secondly, the interaction of the virus with the immune system is very complex, not the least caused by the fact that the docking structure, the ACE2 receptor, is strongly present in many tissues, especially on the endothelial cells. This is a major cause of why the vaccination can provoke unwanted immune reactions (activations) against important life-sustaining systems, as has happened with the clotting system. Knowledge originating from a deep understanding of these mechanisms is very helpful in such cases. It can also help to improve the management of severe COVID-19 and its complications by clotting decompensation and cytokine storms.

## Figures and Tables

**Figure 1 biomedicines-09-01342-f001:**
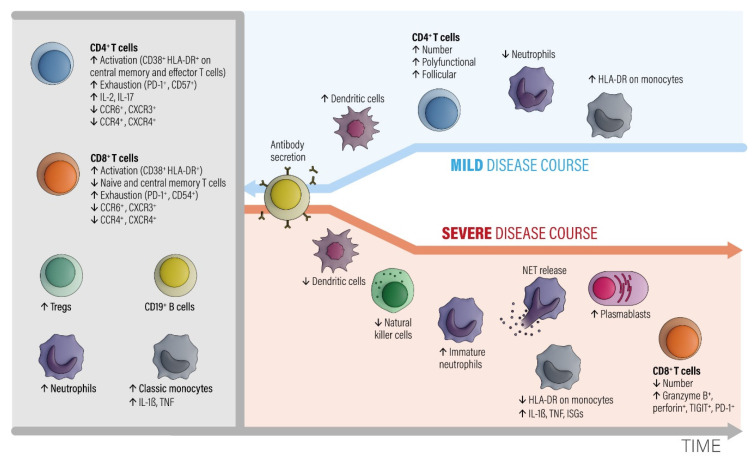
The involvement of different classes of immune cells mediating different courses of COVID-19 during development of the disease, modified from ref. [27]. Abbreviations: CCR = C-C chemokine receptor. CTLA-4 = cytotoxic T-lymphocyte protein 4. CXCR = C-X-C chemokine receptor. HLA-DR = HLA DR isotype. IL = interleukin. ISGs = interferon-stimulated genes. NETs = neutrophil extracellular traps. PD-1 = programmed cell death 1. TNF = tumor necrosis factor. Tregs = regulatory T cells. Modified from [27].

**Figure 2 biomedicines-09-01342-f002:**
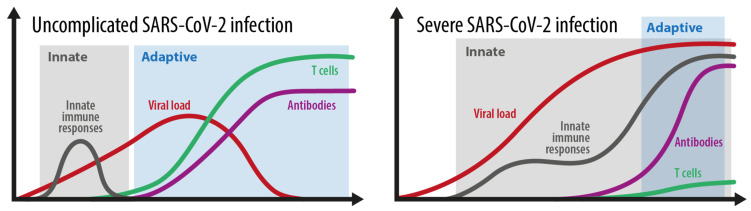
The kinetics of the innate and adaptive immune responses in simple versus severe SARS-CoV-2 infections. Grey area: innate immune response. Blue area: adaptive immune response. Adopted from [52].

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
