# Peer review of "Immune Responses against SARS-CoV-2—Questions and Experiences"

_biomedicines, 2021, doi:10.3390/biomedicines9101342_

Round 1

Reviewer 1 Report

Interesting approach of the subject, but, at the same time, due to a large number of bibliographical references, difficult to verify and follow.

Author Response

We agree that the list of references is rather long. On the other hand, the topic is so dynamic that it is difficult to keep the number of citations low.

Reviewer 2 Report

biomedicines-1358986-peer-review-v1

Even though the author as compiled a lot of points related to SARS-CoV2, there are few changes in the writing that are required.

In abstract, “fully comprehend the complex clinical manifestations, a profound understanding of the immune”, I think work “response” is missing after the word immune.

“Vaccine mediated immunity is much safer than naturally acquired immunity” came abruptly, please provide the context before putting this line.

In introduction, “The innate and the adaptive immune system have to cooperate and coincide”, work “have to” is not necessary.

In Introduction, “if cooperation disturbed”, the author can put more scientific language, such as “Due to the deficiency or over activation of immune response could lead”

In Introduction paragraph, “a bad course. Advanced age [15], cardiovascular disease (CVD) and obesity [16] may play an important role in this” could re-frame work “this” with as “Can also include the genetic factors that could be involved in causing severe COVID-19”

For the line, “Considering the high number of COVID-19 persons in…”, please reframe the sentence.

In heading 1.2, “Preferentially, antibodies in blood a.. “ , should be “Neutralizing antibodies to SARS-CoV-2” in place of antibodies.

Please elaborate in detail what are the immune events involved in innate and adaptive immunity in SARS-CoV2 under heading 1.3.

In heading 1.3 (A), “After entrancing target cells by..”, the word might be “entering”.

Please provide more detail of the adaptive immune response towards the SARS-CoV-2 with workflow diagram.

Please provide approximate percentage of individuals that generate the strong response and cannot transmit, similarly for other scenario under heading 1.4 “Immune responses can vary markedly. Some people..”

Please provide the citation and emphasize on the quality of T-cell under heading 1.4 “he number and quality of the T cell response generated vary markedly between individuals.”

“This statement is counter to the mild symptoms to the COVID as low antibody and lower amount of type I IFNs. Please go in more detail to explain why young individuals has mild symptoms or asymptomatic in comparison to the older individuals” for heading 1.4 “Interestingly, this is a fact although newborns and young children produce lower amounts of type I IFN upon stimulation through the virus and show a reduced breadth of antibody responses to SARS-CoV-2 proteins..”. and reframe the sentence.

Please change “Sars-CoV2” to “SARS-CoV2” wherever required.

In heading 1.5 “…ecovery than those in healthy controls.”, and “…significantly compared with those at the first clinic visit“ Please cite these.

Under heading 1.5, “remains to be be clarified”, “be” is repeated.

Please cite “een sequenced and shown to be different variants of the virus” under heading 1.6.

Under heading 1.8, in my opinion, the immune response or antibody titers are high if the individuals is infected with SARS-CoV-2 naturally and then had a single shor of vaccine. I think authors can include this point also.

Please cite the DOI link under heading 1.8.

For last paragraph in heading 1.10, Please include the SARS-CoV-2 antibody transmission from the mother to the fetus that provide protection to the newborn through vaccine or SARS-CoV-2 naturally infection. Also please include the effect of the COVID-19 infection on the mother and their fetus.

Could you suggest any particular reason, does it is due to any genetic factor that increase the risk of severe COVID-19, in these ancestry for “..more common in Black and Hispanic children” under heading 1.11.

Please cite this line “Perhaps, a longer interval between the vaccination doses promotes a better immune response”

Can focus more on the immune escape variant that could effect the vaccination in heading 2.1.

Please cite “None of the Phase III trials enrolled immunocompromised individuals”.

In heading 2.7, please include the cellular immunity, how much does it last and how it is efficacious.

Please elaborate under heading 2.11, “There is evidence from the AstraZeneca/Oxford vaccine that a longer interval.. “, how long interval are we talking here and please elaborate this point.

Please rewrite the previous statement Undoubtedly the virus has come to stay. Put it in more scientific terms in conclusion.

In conclusion, so if we are considering the COVID-19 at a global scale. In my opinion, this statement is correct. As different point of time, the SARS-CoV-2 infection is prevalent is some or other part of the world.

Author Response

Point to point responses to reviewer 2
Even though the author as compiled a lot of points related to SARS-CoV2, there are few
changes in the writing that are required.
Re.: Thank you.

In abstract, “fully comprehend the complex clinical manifestations, a profound
understanding of the immune”, I think work “response” is missing after the word immune.
Re.: Thank you – we replaced answers by “response”.

“Vaccine mediated immunity is much safer than naturally acquired immunity” came
abruptly, please provide the context before putting this line.
Re.: We provide a context sentence in the revised version (line 33).

In introduction, “The innate and the adaptive immune system have to cooperate and
coincide”, work “have to” is not necessary.
Re.: We erased “have to”.

In Introduction, “if cooperation disturbed”, the author can put more scientific language, such
as “Due to the deficiency or over activation of immune response could lead”
Re.: Corrected (line 47-48).

In Introduction paragraph, “a bad course. Advanced age [15], cardiovascular disease (CVD)
and obesity [16] may play an important role in this” could re-frame work “this” with as “Can
also include the genetic factors that could be involved in causing severe COVID-19”
Re.: Reframed according to the suggestion of the referee (line 74-75).

For the line, “Considering the high number of COVID-19 persons in…”, please reframe the
sentence.
Re.: Reframed (line 85-87).

In heading 1.2, “Preferentially, antibodies in blood a.. “ , should be “Neutralizing antibodies
to SARS-CoV-2” in place of antibodies.
Re.: Replaced (Line 115).

Please elaborate in detail what are the immune events involved in innate and adaptive
immunity in SARS-CoV2 under heading 1.3.
Re.: We improved the headings and added figure 2 in the revised version (line 128, line 164).

In heading 1.3 (A), “After entrancing target cells by..”, the word might be “entering”.
Re.: We replaced Entrancing by “entering (line 130).

Please provide more detail of the adaptive immune response towards the SARS-CoV-2 with
workflow diagram.
Re.: We introduced and commented (line 155-163) a new figure (Figure 2) showing the different
kinetics of the immune response in uncomplicated versus severe COVID-19.

Please provide approximate percentage of individuals that generate the strong response and
cannot transmit, similarly for other scenario under heading 1.4 “Immune responses can vary
markedly. Some people..”
Re.: Unfortunately, I did not find reliable percentage numbers according to this. Nevertheless, I
discuss this fact in the revised version (line 193-203), and added a new reference (54).

Please provide the citation and emphasize on the quality of T-cell under heading 1.4 “he
number and quality of the T cell response generated vary markedly between individuals.”’
Re.: Please refer on the point above.

“This statement is counter to the mild symptoms to the COVID as low antibody and lower
amount of type I IFNs. Please go in more detail to explain why young individuals has mild
symptoms or asymptomatic in comparison to the older individuals” for heading 1.4
“Interestingly, this is a fact although newborns and young children produce lower amounts
of type I IFN upon stimulation through the virus and show a reduced breadth of antibody
responses to SARS-CoV-2 proteins..”. and reframe the sentence.
Re.: We reframed the sentence, and discuss aspects why children are more protected despite weak
type I INF response (line 227-231).

Please change “Sars-CoV2” to “SARS-CoV2” wherever required.
Re.: Done

In heading 1.5 “…ecovery than those in healthy controls.”, and “…significantly compared
with those at the first clinic visit“ Please cite these.
Re.: Cited (line 285).

Under heading 1.5, “remains to be be clarified”, “be” is repeated.
Re.: Corrected.

Please cite “een sequenced and shown to be different variants of the virus” under heading
1.6.
Re.: Cited (line 337).

Under heading 1.8, in my opinion, the immune response or antibody titers are high if the
individuals is infected with SARS-CoV-2 naturally and then had a single shor of vaccine. I
think authors can include this point also.
Re.: Please apologize, I did not find the place in the text where this point fits to be included.
Please cite the DOI link under heading 1.8.

Re.: Sorry, I did not yet find this paper in the PubMed data base. Hence, I can only provide this DOI
link.
For last paragraph in heading 1.10, Please include the SARS-CoV-2 antibody transmission
from the mother to the fetus that provide protection to the newborn through vaccine or
SARS-CoV-2 naturally infection. Also please include the effect of the COVID-19 infection on
the mother and their fetus.
Re.: We add and discuss (line 383-396) two recent papers (Refs 118, 119) commenting on SARS-CoV2 antibody transmission from mother to the newborn after infection on the one hand and
vaccination on the other hand.

Could you suggest any particular reason, does it is due to any genetic factor that increase the
risk of severe COVID-19, in these ancestry for “..more common in Black and Hispanic
children” under heading 1.11.
Re.: To the best of my knowledge, the exact reasons for this remain unknown. Nevertheless, I added
a section discussion some still speculative possible reasons (line 405-415, Refs 122-124).

Please cite this line “Perhaps, a longer interval between the vaccination doses promotes a
better immune response”
Re.: Cited, line 437 (I found only a Medrix org DOI link for this and added it. This paper is not yet in
PubMed).

Can focus more on the immune escape variant that could effect the vaccination in heading
2.1.
Re.: I added a new paragraph focusing this (line 452-465, Refs 132-133).

Please cite “None of the Phase III trials enrolled immunocompromised individuals”.
Re.: Cited

In heading 2.7, please include the cellular immunity, how much does it last and how it is
efficacious.
Re.: In the revised version, we discuss cellular immunity in many details in the text addressing the
new figure 2.

Please elaborate under heading 2.11, “There is evidence from the AstraZeneca/Oxford
vaccine that a longer interval.. “, how long interval are we talking here and please elaborate
this point.
Re.: I added Reference 171 and a sentence (line 646-647)

Please rewrite the previous statement Undoubtedly the virus has come to stay. Put it in
more scientific terms in conclusion.
Re.: I erased this sentence.

This manuscript is a resubmission of an earlier submission. The following is a list of the peer review reports and author responses from that submission.